# Temperature Field of Tool Engaged Cutting Zone for Milling of Titanium Alloy with Ball-End Milling

**DOI:** 10.3390/mi9120672

**Published:** 2018-12-18

**Authors:** Shucai Yang, Chunsheng He, Minli Zheng, Quan Wan, Yuhua Zhang

**Affiliations:** 1College of Mechanical and Power Engineering, Harbin University of Science and Technology, Harbin 150080, China; minli@hrbust.edu.cn (M.Z.); wq@hrbust.edu.cn (Q.W.); 18346073898@163.com (Y.Z.); 2Measurement-Control Technology and Instrument Key Laboratory of Universities in Heilongjiang Province, Harbin University of Science and Technology, Harbin 150080, China

**Keywords:** micro-textured ball-end milling tools, temperature field, heated density function, titanium alloy, thermo-mechanical coupling behavior

## Abstract

When milling titanium alloy, the cutting temperature has a strong impact on the degree of tool wear and, in turn, tool life and the surface quality of the workpiece. The distribution of the temperature field on a tool’s rake face can be improved through the use of micro-textures, which help to reduce friction and, ultimately, wear on the tool. In this paper we present a new way to measure cutting temperature and examine heat distribution when milling titanium alloy with micro-textured ball-end milling tools. We first establish the heat flux density function for the contact area between the workpiece and the tool and then for the rest of the tool. Thermal stress simulation shows that adhesive wear tends to happen in the contact area and on the flank face, rather than at the tip of the tool, with the temperature distribution gradient for the rest of the tool being more uniform. The maximum value for thermal stress on the cutting edge was 2.0782 × 106 Pa. This decrease as you move away from the cutting edge along the contact area between the tool and the workpiece. Maximum deformation of the tool is also mainly concentrated at the principal contact point, with a value of 1.9445 × 10^−9^ m. This, too, decreases as you move away from the cutting edge and into the rest of the contact area. This research provides the basis for the optimization of tool structure and further investigation of the thermo-mechanical coupling behavior of micro-textured ball-end milling cutters when milling titanium alloy.

## 1. Introduction

In titanium alloy milling processes, a key cause of cutting tool failure is adhesion-based wear on cutting tool surfaces. The degree of adhesion is heavily influenced by the heat generated during cutting. Obviously, this also affects tool service life and workpiece surface quality [1,2]. When a micro-texture is applied to the rake face of a ball-end milling tool, it can help to lower the temperature by reducing friction. This makes it important to examine the exact heat transfer mechanisms in play when using such tools and to establish certain key measures, such as the heat density function at the rake face. By solving this and revealing heat-related damage mechanisms it is possible to provide a theoretical basis for optimizing the geometric parameters associated with tool design, thus improving cutting efficiency, the quality of worked surfaces and cutting tool service life.

There are two main ways to approach the study of cutting temperature: Analytically and experimentally. Analytic methods involve establishing a simplified analytic model. Experimental methods usually involve various approaches to measurement. For cutting temperature, these approaches may include the use of thermocouples, metallographic separation or infrared thermal imaging [3,4,5]. Thermocouples have many advantages. They can measure a wide range of temperatures, are strong, durable and responsive and are, themselves, very resistant to heat. As a result, thermocouples are the most commonly used form of measurement. However, getting a good signal from thermocouples can be complex and improper handling will lead to errors and low precision. They are also subject to corrosion. This being the case, a new method for measuring cutting temperatures is required.

Jaeger [6] has extensively researched cutting heat and has put forward the hypothesis that heat arising from friction between the cutting tool and the workpiece is evenly distributed. This is of foundational significance for the theoretical calculation of cutting temperature. On the basis of this hypothesis, it is possible to use mathematical equations and heat transfer formulae to calculate the temperature of the shear and rake faces. Jaeger used his observations to deduce formulae for the average temperature distribution across friction surfaces. However, this concentrates on two-dimensional cutting at the rake face. Zhou [7] has done a lot of work on the calculation and distribution of temperature in the contact area between the cutting tool and the workpiece and has proposed an analysis that differs from Jaeger’s. In his view, the intensity with which friction is distributed between cutting tools and workpieces is usually uneven. This has a significant impact on friction-induced temperature. Chakrverti and Hoshi [8] studied the damage mechanisms related to interrupted cutting, including thermal stress and thermal fatigue and found thermally-induced cracking on the rake face, with some cracks being perpendicular to the main cutting edge and others parallel to it. Guo et al. [9] have also looked at thermal stress and the damage caused by milling temperatures. In their case, they used finite element analysis to calculate the instantaneous milling temperature at the cutting edge. They also derived the start and boundary conditions for heat conduction. On the basis of this, they established a method for calculating fluctuating temperature fields and theoretically deduced the temperature distribution on the contact surface at the point where the tool cuts into and out of the workpiece. This has had a very important influence on studies of tool life and tool damage. Wang et al. [10] have focused on the relationship between cutting temperature, different blades shapes and failure caused by adhesion. They found that the cutting temperature for wave-edged milling blades and blades with a large front angle is notably lower than it is for flat milling blades, thus improving their ability to resist adhesion-based failure. Xia [11], has examined the impact of different factors on cutting temperature and emphasizes the importance of three in particular. With regard to the workpiece material, it was noted that, when it is easy to cut, it has little effect on the cutting temperature. However, as its hardness increases, the influence of the cutting parameters on the cutting temperature becomes very obvious. Using finite element simulation, Ren et al. [12] have developed empirical formulae for the average cutting temperature of end-milling tools and the relationship between their cutting parameters and the cutting temperature. Xu [2], meanwhile, has established a mathematical model for monitoring tool wear based on cutting temperature. This model is derived from the variance value of the distribution density function of the cutting temperature and qualitative analysis of the relationship between an auto-correlation function of the cutting temperature, the probability distribution and actual tool wear. The modeling of cutter’s displacements during ball end milling with various surface inclinations was established. The cutter’s displacements (vibrations) model including: Tool’s geometry, cutting conditions, surface inclination angle, run out and tool’s deflections (induced by the cutting forces) was proposed. Experiments were carried out with the application of laser displacement sensor and force dynamometer. The research revealed that cutter’s displacements are strongly affected by the cutter’s run out and surface inclination. This observation is also confirmed by the developed model [13]. A method for the reduction of forces and the improvement of efficiency during finish ball end milling of hardened 55NiCrMoV6 steel was proposed. The primary objective of this work concentrates on the optimal selection of milling parameters (cutting speed (*v_c_*), surface inclination angle *α*), which enables the simultaneous minimization of cutting force values and increased process efficiency. This procedure is carried out with the application of the response surface method, based on the minimization of a total utility function. The work shows that the surface inclination angle has a significant influence on the cutting force values. Minimal cutting forces and relatively high efficiency can be achieved with cutting speed *v_c_* = 375 m/min and surface inclination angle *α* = 15° [14]. An analysis of relations between the instantaneous tool displacements and surface roughness formed during ball end milling of the surface with the inclination towards the tool’s axis was present. A novel experimental method for the estimation of ball end mill’s working part vibrations, considering displacements correlated with the geometrical errors of the tool holder-spindle system and deflections caused by milling forces has been proposed. The experiments have been conducted on hard-to-cut low carbon hardened alloy steel. The investigations show that the value of tool’s overhang significantly affects the mechanisms of surface roughness generation during finishing ball end milling. In the case of milling with the rigid tool (*l* = 35 mm), the surface roughness is strongly correlated with the kinematic-geometric model, as well as with geometrical errors of the machining system. Nevertheless, in case of milling with the slender tool (*l* = 85 mm), the surface roughness formation is mainly affected by the tool’s working part dynamic deflections caused by milling forces [15].

To sum up, many scholars have studied the temperature distribution for cutting tools. However, most traditional methods for measuring the cutting temperature have disadvantages. There has been less research regarding the relationship between heat and micro-textured milling tools. We have therefore developed a new experimental method to tackle these omissions. Looking specifically at the interaction between micro-textured ball-end milling tools and cutting temperature when milling titanium alloy, we first analyzed the heat source to establish a heat flow density function for the tool-workpiece contact area. We then used the heat density function to develop a temperature field model. After this, we conducted a simulation analysis of the temperature field using the ANSYS Workbench and compared the results to the theoretical model. The results of this analysis can be used to inform the future optimization of cutting tool structures and provide useful input for the ongoing study of the thermo-mechanical coupling behavior between cutting tools and difficult to cut materials.

## 2. Experimental Study

### 2.1. Aims

To begin our study, we wanted to look at how milling temperature changes over time. To do this we set up an experimental platform. A single factor milling experiment was carried out where various cutting parameters (cutting speed, cutting depth, and feed) were systematically changed whilst the temperature was being measured. Changes in the values of the milling temperature during the milling cycle in relation to changes in the parameters provided the basic data for the next part of the study, which was the development of a heat density function. The specific objective here was to arrive at a principled way of calculating the relationship between heat and the cutting cycle. With this in hand we were able to simulate the heating of the rake face of a micro-textured ball-end milling tool that we could compare to the experimental data and the derived theory.

### 2.2. Equipment and Procedure

For the workpiece we used titanium alloy TC4. A ball-end milling tool was used for the cutting. As the tool nose is involved in cutting all of the time with this kind of cutter, the line speed is always zero, making the tool point wear faster and reducing its life. The presence of wear, in turn, affects the quality of the machined surface. We wanted to engage in using a cutting technique under robust conditions so that we could realistically assess the impact of the micro-textured surface on the tool. Note that the workpiece was given an inclined angle was 15°, because some research suggests that this machining angle provides the best cutting performance [16]. The overall machining method was progressive milling.

The experiment used a three axis VDL-1000E numerically-controlled milling tool for the machining. A Kistler 9257B dynamometer was used to measure the milling force during the process. The temperature was obtained using a NANMAC E12-3-K-U thermocouple. This type K thermocouple is specifically designed so that it can be used on machines of any shape. It was therefore possible to design and build a special fixture that could be attached to the thermocouple, as well as the dynamometer and the workpiece. The approach to measuring the milling temperature is shown in Figure 1.

We adopted a single factor approach for the experiment. In other words, the experiment was designed around single changes in the cutting parameters, as shown in Table 1. We drilled into the center of the workpiece so that the thermocouple could be used to measure changes in the milling temperature over time for each set of cutting parameters. One set of parameters was selected for each milling experiment. The basic cutting speed was 120 m/min, the basic cutting depth was 0.7 mm and the basic feed was 0.08 mm/z.

As milling processes are discontinuous rather than steady, the temperature field for a milling tool will always vary over time. In micro-textured ball-end milling processes, there is shear slippage of the cut out chip in the first deformation zone and there is extrusion and friction between the chip, cutting tool and the workpiece in the second and third deformation zones. This causes a rapid rise in temperature. As the chip is cut-out, the temperature of the blade drops quickly. These phenomena mean that cutting tools have to withstand a constantly changing temperature field and this is why an E12-3-K-U type thermocouple had to be used to measure the milling temperature. Changes in the milling temperature over time were measured in relation to changes in the cutting speed and feed rate. Variation curves for these measurements are shown in Figure 2 and Figure 3, respectively.

We used an origin to draw the curve point by point so as to be able to analyze variations in the milling temperature as the cutting speed and feed rate per tooth was changed. The temperature value selected for each cutting parameter was the peak value collected by the experiment. It can be seen from Figure 4 that the cutting temperature increases with an increase in the cutting speed and feed rate per tooth. The increase in temperature associated with the cutting speed is especially noticeable. The is because, as the chip flows out along the rake face, there is a strong friction between the rake face and the bottom of the chip, which produces a lot of heat. This friction-induced heat is mainly concentrated at the very bottom of the chip, but some of it flows up to the top surface of the chip and on into the cutting tool. Therefore, an increase in the cutting speed results in an increase in friction heat, which leads to an increase in the internal temperature of the cutting tool. An increase in the feed rate per tooth also leads to a rise in temperature, but the magnitude is less significant.

Changes in the milling temperature across the milling cycle are shown in Table 2.

The collected data for the milling temperature was fitted using MATLAB. The fitting function for changes in temperature over time is visualized in Figure 5.

## 3. Calculation of the Heat Flux Density Function for the Contact Region between the Chip and the Workpiece

### 3.1. Analysis of the Heat Source 

Cutting heat is generated by the energy consumed during cutting, which is caused, in turn, by the elastic and plastic deformation of the metal in the cutting layer and friction between the rake face, the flank face, the workpiece and the chip [17]. As shown in Figure 6, the heat generated by cutting metal has three aspects: The heat generated by deformation in the shear zone *Q_s_*; the heat *Q_rf_* produced by friction between the chip and the rake face; and the heat *Q_αf_* produced by friction between the machined surface and the flank face. These three heat zones correspond to the three deformation zones, with heat being conducted through the chip, the workpiece, the rake face and the flank face. In the shear zone, part of the cutting heat is transferred to the workpiece and the rest is dissipated with the chip as it flows out. In the contact area, part of the heat is transferred to the tool and the rest is dissipated with the chip. There is also some heat distribution between the cutting tool and the workpiece.

In Figure 6, the heat in the shear zone conducted to the chip and the workpiece is
(1)$$Q_{jc}=R_{1}q_{s},$$
(2)$$Q_{jw}=\left(1-R_{1}\right)q_{s},$$
where *R*_1_ represents the ratio of the heat in the shear zone conducted to the chip and *q_s_* represents the amount of heat on the shear surface per unit time and unit area.

Similarly, the heat conducted into the chip and the cutting tool that is generated in their contact area and the heat conducted into the workpiece and the cutting tool that is generated in their contact area, can be expressed through the following equations:(3)$$Q_{qc}=R_{2}q_{r},$$
(4)$$Q_{qw}=\left(1-R_{2}\right)q_{r},$$
(5)$$Q_{hc}=R_{3}q_{w},$$
(6)$$Q_{hw}=\left(1-R_{3}\right)q_{w},$$
where *q_r_* is the heat generated per unit time and unit area on the rake face and *q_w_* is the heat generated per unit time and unit area on the flank face. *R*_2_ represents the ratio of heat on the rake face conducted into the chip and *R*_3_ represents the ratio of heat on the flank face conducted into the workpiece.

The total cutting heat *Q* can be calculated using: The heat *Q_ch_* conducted into the chip; the heat *Q_c_* conducted into the cutting tool; the heat *Q_w_* conducted into the workpiece; and the heat *Qr* absorbed by the surrounding medium. The heat transfer conduction relationship thus be expressed as:
(7)$$Q_{s}+Q_{rf}+Q_{af}=Q_{ch}+Q_{c}+Q_{w}+Q_{r}.$$

When cutting plastic metals, the cutting heat is mainly generated by deformation in the shear zone and friction on the rake face. When cutting brittle metals, the proportion of friction is larger. Titanium alloy is a plastic metal, so cutting heat is mainly conducted in the shear zone and the contact area between the chip and the tool. For the cutting of titanium alloys, the overall temperature can be calculated on the basis of deformation of the cutting area, friction and heat conduction.

### 3.2. The Average Temperature in the Contact Area 

To solve the heat density function for a micro-textured ball-end milling tool you first of all need to solve the cutting temperature per unit area on the rake face. Traditionally this is done by solving the ratio of the heat on the rake face that is conducted into the chip in relation to the total heat of the rake face. The total heat can be solved using forces in three directions. However, for the machining of titanium alloy with micro-textured ball-end milling tools this is an extremely complex and cumbersome process and the differential equations are hard to establish. This being the case, we used dimensional analysis instead. Dimensional analysis, also known as factor analysis, assumes that dimensions are homogeneous and uses this to uncover the relationship between each physical quantity. This simplifies the test results and makes them easier to collate.

To solve the heat density function using dimensional analysis we proceeded as follows: The cutting speed *v_c_*, the cutting depth *a_c_*, the cutting energy *u*, the heat capacity of the volume *ρc*, and the thermal conductivity coefficient *λ* of the workpiece material, were deemed to be the main factors affecting the average temperature *θ* in the contact area between the cutting tool and the chip. In order to simplify the calculation, *ρc* and *λ* were treated as whole constants. The relationship between the variables that affect the heat density function can thus be expressed as:(8)$$f\left(\theta ,u,\rho c\lambda ,v_{c},a_{c}\right)=0.$$

The relationship between the average temperature *θ* in the tool-chip contact area and the various factors is:(9)$$\theta =pu^{x}{\left(\lambda \rho c\right)}^{y}v_{c}{}^{z}a_{c}{}^{w},$$
where *p* is a coefficient.

The variables and associated dimensions for each factor are shown in Table 3.

The dimensional expression of Equation (9) is
(10)$$\left[\bar{\theta }\right]={\left[ML^{-1}t^{-2}\right]}^{x}{\left[M^{2}t^{-5}{\bar{\theta }}^{-2}\right]}^{y}{\left[Lt^{-1}\right]}^{z}{\left[L\right]}^{w}.$$

As the left and right sides of Equation (10) are equal, we can obtain the following:(11)$$\{\begin{array}{c} x+2y=0\\ -x+z+w=0\\ -2x-5y-z=0\\ -2y=1 \end{array}.$$

From Equation (11) we get *x* = 1, *y* = –1/2, *z* = 1/2, *w* = 1/2. So, the relationship between the average temperature in the tool-chip contact area and each of the influencing factors can now be expressed as
(12)$$\bar{\theta }=pu{\left(\frac{v_{c}a_{c}}{\lambda \rho c}\right)}^{\frac{1}{2}}.$$

The specific cutting energy relates to the power involved in cutting a chip out of a workpiece’s volume. The main cutting force is exerted in the direction of the feed rate. If we assume that the total cutting area is *A*_1_, and the power over a certain unit of time is *W_z_*, then the specific cutting energy *u* can be expressed as
(13)$$u=\frac{W_{z}}{v_{c}\cdot A_{1}}=\frac{F_{x}}{A_{1}}.$$

From previous work we know that the relevant formulae for cutting force in the feed direction for a micro-textured ball-end milling tool are:(14)$$F_{x}=987.14a_{c}{}^{0.2841}\cdot a_{f}{}^{0.3768}\cdot K,$$
(15)$$K={\left(\frac{\sigma_{b}}{638}\right)}^{0.3},$$
where *K* is the correction coefficient for the workpiece material and *σ_b_* is its tensile strength (Mpa).

When cutting titanium alloy, the shape of the workpiece results in a cutting width *a_e_* and a cutting thickness *a_c_* that vary over cutting time, so the relationship between them and the cutting area *A*_1_ is
(16)$$A_{1}=\sum_{1}^{z}a_{c}\cdot a_{e},$$
where *z* is the number of teeth. It can be seen from this formula that the cutting area *A*_1_ also varies over time. In order to simplify the calculation, we can use the average cutting area *A_av_* and the average cutting thickness *a_cav_* instead, which gives the following:(17)$$A_{av}=\frac{af\cdot ac\cdot ae\cdot z}{\pi \cdot D},$$
(18)$$a_{cav}=a_{f}\cdot \sqrt{\frac{a_{c}}{D}},$$
where *D* is the diameter of the tool in mm. The specific cutting energy u can now be expressed as:(19)$$u=\frac{987.14a_{c}{}^{0.2841}\cdot a_{f}{}^{0.3768}\cdot K\cdot \pi D}{a_{f}\cdot a_{c}\cdot a_{e}\cdot z}.$$

As milling here amounts to milling with a single tooth, Equation (19) can be simplified as follows:(20)$$u=493.57\pi \cdot a_{c}{}^{-1.7159}\cdot a_{f}{}^{-0.6232}\cdot K.$$

The average temperature of the tool-chip contact area can now be expressed as:(21)$$\theta =493.57\pi \cdot a_{c}{}^{-1.7159}\cdot a_{f}{}^{-0.6232}\cdot K\cdot p\cdot {\left(\frac{v_{c}\cdot a_{f}}{\lambda \rho c}\cdot \sqrt{\frac{a_{c}}{D}}\right)}^{\frac{1}{2}}.$$

Collating the formulae:(22)$$\theta =493.57\pi \cdot a_{c}{}^{-1.4695}\cdot a_{f}{}^{-0.1232}\cdot K\cdot p\cdot {\left(\frac{v_{c}}{\lambda \rho c}\right)}^{\frac{1}{2}}\cdot D^{-0.25}.$$

As the units for the variables in this dimension are different from those in actual processing, unified units are needed. The units for each variable in the tool-chip contact area in this dimension are shown in Table 4.

In actual processing, the unit for *a_c_* is mm, and the unit for *a_f_* is mm/z. In that case, the average temperature *θ* of the tool-chip contact area can be converted as follows:(23)$$\theta =493.57\pi \cdot {\left(1000a_{c}\right)}^{-1.4695}\cdot {\left(1000a_{f}\right)}^{-0.1232}\cdot K\cdot p\cdot {\left(\frac{vc}{\lambda \rho c}\right)}^{\frac{1}{2}}\cdot D^{-0.25}.$$

The average temperature *θ* of the tool-chip contact area can now be calculated using Table 4:(24)θ=2.288p.

The dimensional units for the parameters relating to both the cutting process and the workpiece material are shown in Table 5.

By using the experimental data regarding the cutting temperature on the rake face of the cutting tool, changes in the average temperature *θ* over time in the tool-chip contact area can be obtained using MATLAB.
(25)$$\theta =\theta_{1}-\theta_{0}=-3.4328\times 10^{15}t^{5}+3.2558\times 10^{13}t^{4}-1.0966\times 10^{11}t^{3}\;+1.5000\times 10^{8}t^{2}-3.2772\times 10^{4}t+23.7501,$$
where *θ*_0_ is the initial room temperature. From Equations (24) and (25), the undetermined coefficient *p* can now be obtained:(26)$$p=-1.5003\times 10^{15}t^{5}+1.4230\times 10^{13}t^{4}-4.7928\times 10^{10}t^{3}\;+6.5559\times 10^{7}t^{2}-1.4323\times 10^{4}t+10.3802,$$
(27)$$\theta =P\cdot \pi \cdot a_{c}{}^{-1.4695}\cdot a_{f}{}^{-0.1232}\cdot K\cdot {\left(\frac{v_{c}}{\lambda \rho c}\right)}^{\frac{1}{2}}\cdot D^{-0.25},$$
where *P* = −1.2340 × 1013*t*^5^ + 1.1707 × 1011*t*^4^ − 3.9430 × 108*t*^3^ + 5.3935 × 105*t*^2^ − 1.1783 × 102*t* + 0.0854.

### 3.3. Calculation of the Heat Flux Density Function 

Friction between the chip and the rake face results in a reduction of the cutting speed, which leads to secondary slippage. The speed in the tool-chip contact area is significantly reduced near the tool edge, but gradually increases as it moves away. The thickness of the contact layer decreases as the speed increases, until it reaches a certain value. Then a retention layer is formed at the bottom of the chip. Secondary slippage causes deformation of the chip in the shear plane, thus consuming part of the energy. In the tool-chip contact area, friction between the chip and the rake face also consumes part of the energy. The sum of these two energies is then the total power of the chip as it flows out from the contact area. So, in general, cutting heat in the chip and retention layer is generated by the flow of the chips. The area of the retention layer is much smaller than that of the tool-chip contact area. So, this heat source can be regarded as a limited surface heat source. The cutting conditions and material properties of the workpiece have a significant influence on the average thickness of the retention layer, which is linearly related to the contact length between the tool and the chip:(28)$$\bar{h}=\varepsilon \cdot l_{f}.$$

According to the average temperature of the tool-chip contact area, the total heat flux in the contact zone will be:(29)$$H=c\rho l_{f}l_{w}\bar{h}\cdot \theta .$$

The heat density function is the total heat flux per unit area and the heat flux density in the tool-chip contact area is
(30)$$\bar{q}=\frac{H}{l_{f}l_{w}}=\varepsilon c\rho l_{f}\theta ,$$
where *ε* is the average thickness of the retention layer, which is a constant that varies according to the cutting conditions and properties of the material. The expression of the contact length *l_f_* can be obtained as follows:(31)$$l_{f}=1.219a_{f}+0.512.$$

Combining Equations (27), (30) and (31), the heat flux density for the tool-chip contact area is thus:(32)$$\bar{q}=P\cdot \pi \cdot a_{c}{}^{-1.4695}\cdot a_{f}{}^{-0.1232}\cdot K\cdot {\left(\frac{v_{c}\rho c}{\lambda }\right)}^{\frac{1}{2}}\cdot D^{-0.25}\cdot \left(0.3726a_{f}+0.1690\right).$$

### 3.4. The Heat Destiny Function 

If we ignore any power in the third deformation zone and focus just on the cutting power for titanium alloy consumed by the first and second deformation zones, the temperature field for the tool-chip contact area can be worked out from the known heat sources. The boundary conditions here can be solved using differential equations for heat conduction. However, to solve the heat density function for the tool-chip contact area you do not use the temperature in the zone, but the heat coming from the heat sources. Solving the temperature of the heat sources involves solving a series of unknown quantities, so mathematical analysis and numerical methods are not applicable. That being the case, it is not worth trying to apply a differential equation for heat conduction to solve the boundary conditions. Instead, the heat sources for the first and the second deformation zones are taken to be equivalent to the one on the planar. The area is then taken to be equivalent to a semi-infinite or quarter-infinite object. For this, it is possible to use complex field analysis. This method is applied to the temperature fields of unsteady heat sources, i.e., when the temperature is changing over time. From this, the relationship between the temperature field and its influencing factors can be established.

In this paper, we assume that the size of the heat source is limited and that it is in the center of the object, with the heat being conducted in any direction being almost the same. To simplify the boundary conditions, here we will be assuming that the object is a semi-infinite object. The surface heat source method can then be used to solve the boundary conditions of the temperature field in the tool-chip contact area.

According to Equation (32), we know that the heat source intensity per unit time for the surface heat source in the tool-chip contact area is *Q_m_*. Its contact width is *l_w_* and its contact length is *l_f_*, as shown in Figure 7.

The distance between any strip of points within the heat source and the original point at which heating occurs is dyi. So, the differential form of temperature rise in the heat source can be calculated as follows:(33)$$d\theta =\frac{Q_{m}dy_{i}}{\pi c\rho at}e^{-\frac{{\left(y-y_{i}\right)}^{2}+z^{2}}{4at}}\left[erf\left(\frac{x}{\sqrt{4at}}\right)-erf\left(\frac{x-l_{w}}{\sqrt{4at}}\right)\right].$$

The total temperature rises for a point M4 (*x*_4_, *y*_4_, *z*_4_) within the entire surface heat source is
(34)$$\theta =\frac{Q_{m}}{c\rho \left(\pi at\right)}e^{-\frac{z_{4}{}^{2}}{4at}}\left[erf\left(\frac{x_{4}}{\sqrt{4at}}\right)-erf\left(\frac{x_{4}-l_{w}}{\sqrt{4at}}\right)\right]\overset{l_{f}}{\underset{0}{\int }}e^{-\frac{\left(y_{4}-y_{i}\right)}{4at}}dy_{i}.$$

Collating the equations:(35)$$\theta =\frac{Q_{m}}{c\rho {\left(\pi at\right)}^{1/2}}e^{-\frac{z_{4}{}^{2}}{4at}}\left[erf\left(\frac{x_{4}}{\sqrt{4at}}\right)-erf\left(\frac{x_{4}-l_{f}}{\sqrt{4at}}\right)\right]\left[erf\left(\frac{y_{4}}{\sqrt{4at}}\right)-erf\left(\frac{y_{4}-l_{w}}{\sqrt{4at}}\right)\right]$$

Equation (35) is the temperature field distribution model for the total heat on the rake face. The heat intensity for the tool-chip contact area is:(36)$$Q_{m}=q_{ft}.$$

Using Equations (35) and (36), the temperature at any point on any surface of the cutting tool can be calculated thus:(37)$$\theta =\frac{\bar{q}}{c\rho {\left(\pi at\right)}^{1/2}}e^{-\frac{z_{4}{}^{2}}{4at}}\left[erf\left(\frac{x_{4}}{\sqrt{4at}}\right)-erf\left(\frac{x_{4}-l_{f}}{\sqrt{4at}}\right)\right]\left[erf\left(\frac{y_{4}}{\sqrt{4at}}\right)-erf\left(\frac{y_{4}-l_{w}}{\sqrt{4at}}\right)\right].$$

According to the previous equations, as point M4 (*x*_4_, *y*_4_, *z*_4_) is located on the rake face, so *z*_4_ = 0. In that case, the temperature field model can be transformed into a heat density function for the rake face as follows:(38)$$\theta =\frac{\bar{q}}{c\rho {\left(\pi at\right)}^{1/2}}\left[erf\left(\frac{x_{4}}{\sqrt{4at}}\right)-erf\left(\frac{x_{4}-l_{f}}{\sqrt{4at}}\right)\right]\left[erf\left(\frac{y_{4}}{\sqrt{4at}}\right)-erf\left(\frac{y_{4}-l_{w}}{\sqrt{4at}}\right)\right].$$

## 4. Simulation of the Temperature Field 

### 4.1. Establishing the Simulation Model 

As noted above, the cutting temperature generated during cutting determines the degree of adhesion on the surface of the cutting tool. This will affect its service life and the surface quality of the workpiece. In that case, it is important to analyze the temperature distribution across the tools involved in the cutting. However, as these kinds of experiments can be expensive, finite element simulation is usually used instead [18]. We therefore sought to optimize the texture and geometric parameters for micro-textured ball-end milling tools using a finite element simulation. This also gave us further insights regarding tool-workpiece thermo-mechanical coupling behavior. As the milling temperature changes over time during the milling process, the temperature field was treated as an instantaneous temperature field for the purposes of simulation.

The temperature distribution for a micro-textured ball-end milling cutter was simulated by an ANSYS Workbench and SolidWorks was used for the modeling. A comprehensive simulation of the tool would have been extremely complex and the modelling would therefore have taken too long, so the model had to be simplified. As the micro-texture is only located in the contact area between the cutter and the chip on the rake face, the simulation was only carried out for the temperature of the blade. A basic model of the blade is shown in Figure 8. It is an indexable blade with a diameter of 20 mm. The rake face micro-texture consists of micro-pits with a diameter of 50 μm and a depth of 35 μm. The distance between the first row of micro-pits and the cutting edge is 120 μm, as is the distance between any two adjacent micro-pits. The material parameters for the tool are shown in Table 6.

After modeling, a temperature distribution simulation was carried out. The constitutive steps for this were: Inputting the model; definition of the material attributes; partitioning of the meshes; definition of the boundary conditions; analysis; solution; and image analysis. The generation of the meshes was the most important part. A uniform mesh ensures that the temperature distribution will also be uniform. When refining the meshes, especially those in the temperature distribution area, we found that most could be divided into tetrahedrons. As the majority related to second-order elements, the initial mesh quality was poor. The accuracy and speed of calculation can be affected by the simulation quality of these kinds of meshes, so their optimization was essential. The optimized meshes are shown in Figure 9a.

The mesh optimization was performed using the ICEM CFD module in the ANSYS Workbench. This allows for CAD model restoration, automatic middle surface extraction, unique mesh sculpturing, editing, and solver support. There are three kinds of ICEM mesh models: A hexahedral mesh; a tetrahedral mesh; and a prismatic mesh. The hexahedral mesh can generate multi-extensible block structures and an unstructured mesh. The tetrahedral mesh is the one best-suited to the fast and efficient meshing of complex models, because the mesh generation is automated. The prismatic mesh is mainly used for refining the boundary layer, or transitioning between meshes of different shapes. Compared to the tetrahedral mesh, the prismatic mesh is more regular and can provide better computational regions at the boundary. In our study, an important constraint was that the micro-texture was very small. As stress was obviously going to be concentrated in the contact area where the micro-texture was located, the meshes needed to be finely divided. In order to avoid excessive calculation time, optimized tetrahedral meshes therefore had to be used. However, the meshes could be larger outside of the contact area, so long as the transition from the smaller units to the larger ones was smooth. After division, the mesh accuracy was checked and any distorted meshes were modified. The optimized mesh is shown in Figure 9b. It can be seen that the optimized mesh nodes were finer, making the calculation more accurate and rapid. When the simulation was complete, it was found that the temperature distribution closely matched the distribution found in actual machining.

### 4.2. The Boundary Conditions and Loads

Before determining the boundary conditions for thermal analysis, we needed to analyze the way heat is conducted through the temperature field from the heat source. Heat can be conducted in three principle ways: Heat conduction; heat convection; and heat radiation. Heat conduction can be defined as the exchange of internal energy caused by a temperature gradient between two parts of a body that are completely in contact with each other, or between different parts of an object. Heat conduction follows Fourier’s law, which is:(39)$$q^{n}=-k\frac{dT}{dx},$$
where *q^n^* represents the density of the heat flow rate, and *k* represents thermal conductivity. A minus sign represents decreasing temperatures.

Heat convection is the exchange of heat between the surface of a solid and the medium that is in contact with it, because of a temperature difference. Heat convection can be divided into two categories: Natural convection and forced convection. Heat convection is described by the Newtonian cooling equation,
(40)$$q^{n}=h\left(T_{S}-T_{B}\right),$$
where *h* represents the convective heat transfer coefficient, *T_S_* represents the temperature of the solid surface, and *T_B_* represents the temperature of the surrounding medium.

Thermal radiation refers to the process of exchange whereby an object emits electromagnetic energy and it is absorbed by other objects and translated into the heat. The higher the temperature of an object, the more heat it will radiate per unit time. Heat conduction and heat convection require a medium for transferring heat, whilst heat radiation does not. Heat radiation is at its most efficient in a vacuum. It can be described as follows:(41)$$\frac{\partial \theta }{\partial t}=\frac{\lambda }{c\cdot \rho }\left(\frac{\partial^{2}\theta }{\partial x^{2}}+\frac{\partial^{2}\theta }{\partial y^{2}}+\frac{\partial^{2}\theta }{\partial z^{2}}\right),$$

According to the principle of the conservation of energy, variations in a temperature field can be obtained by solving the differential equation for heat conduction. However, to solve this, initial and boundary conditions need to be satisfied. Solving the initial condition involves knowing the temperature distribution of the whole body at the beginning of the heat conduction process. It can be expressed as:(42)$$\{\begin{array}{l} {\left(t\right)}_{\tau =0}=t_{0}\\ {\left(t\right)}_{\tau =0}=f\left(x,y,z\right) \end{array},$$
where *t*_0_ represents the initial temperature of the object, with it being assumed that the temperature of the object is uniform at the outset. *f*(*x*, *y*, *z*) represents a function that indicates that the initial temperature of the object will vary according to the coordinates *x*, *y*, and *z*.

The boundary conditions relate to the heat conduction between the boundary of the object and the surrounding medium after the beginning of the process. These boundary conditions can be divided up as follows:

Boundary condition (1): The functional relationship of the temperature (*t*)_Γ_ that varies according to the position and time at each known point. In the simplest case:(43)$$\{\begin{array}{l} {\left(t\right)}_{\Gamma }=f\left(x,y,z,\tau \right)\\ {\left(t\right)}_{\Gamma }=t_{w} \end{array},$$
where *f*(*x*, *y*, *z*, *τ*) is the functional relationship between the known boundary temperature and various positions and times; *t_w_* is the boundary temperature, which has a constant value; and the *Γ* in subscript represents the boundary.

Boundary condition (2): The functional relationship of the heat flux density that varies at each known point in position and time. Under conditions of stable heat conduction (*q_n_*)_Γ_ = *q_w_*. Thus,
(44)$$\{\begin{array}{l} {\left(q_{n}\right)}_{\Gamma }=-\lambda {\left(\frac{\partial t}{\partial n}\right)}_{\Gamma }=f\left(x,y,z,\tau \right)\\ {\left(q_{n}\right)}_{\Gamma }=-\lambda {\left(\frac{\partial t}{\partial n}\right)}_{\Gamma }=q_{w} \end{array},$$
where ${\left(q_{n}\right)}_{\Gamma }=-\lambda {\left(\frac{\partial t}{\partial n}\right)}_{\Gamma }$ is the vector expressing Fourier’s law; *n* is the normal at any point on the boundary; *f*(*x*, *y*, *z*, *τ*) is the functional relation between the heat flux density and some known point in position and time; and *q_w_* is the heat flux density of the boundary.

If the boundary is an adiabatic boundary, there is no heat conduction and the heat flux density is zero, giving:(45)$${\left(\frac{\partial t}{\partial n}\right)}_{\Gamma }=0.$$

Boundary condition (3): The temperature of the surrounding medium tf and the heat release coefficient *α*, known at any point where there is convection heat conduction between the surface of the object and the medium. According to the principle of the conservation of energy, the heat exchange between the surrounding medium and the heat conduction surface of an object should be equal to the heat conducted from the object to the surface per unit time. Thus,
(46)$$\theta =pu^{x}{\left(\lambda \rho c\right)}^{y}v_{c}{}^{z}a_{c}{}^{w},$$
where *α* is the exothermic coefficient, which is determined by the physical state of the medium and *t_f_* is the temperature of the medium.

Research studying the actual nature of temperature fields frequently uses the above three boundary conditions. The third, in particular, is often encountered in engineering. Taken together, the differential equation for heat conduction and its boundary conditions forms a complete mathematical model for solving the problem of heat conduction. In the process of milling titanium alloy with micro-textured ball-end milling tools, the tool-chip contact area can be considered to be the principal heat source. On that basis, finite element simulation analysis can be used to solve the temperature field for the cutting tool according to the above three boundary conditions.

Milling titanium alloy with a micro-textured ball-end milling tool is what is known as a ‘cut-in’ and ‘cut-out’ process. In that case, the temperature of the cutting tool will rise initially as it cuts in, then decrease as it cuts out. A further point to note is that, in actual practice, the process involves the use of a single tooth when cutting. So, when analyzing the temperature field of the heat source, only the heat source for one tooth needs to be considered. The overall temperature of the cutting tool was 28 °C at room temperature before analysis. The geometric model of the cutting tool is shown in Figure 10. The cutting tool was installed in the tool bar. Surface C, the back face C1, and face D were in close contact with the tool bar. Screws were then fixed through the center hole, and face I was in close alignment with the screws. Surfaces B, E, F, G and H faced the air, and surface A was the tool-chip contact area. When milling, surface A was treated as the principal heat source for the temperature field.

The boundary conditions for the temperature field were:(47)$$\{\begin{array}{l} \frac{\partial \theta }{\partial t}=\frac{\lambda }{c\cdot \rho }(\frac{\partial^{2}\theta }{\partial x^{2}}+\frac{\partial^{2}\theta }{\partial y^{2}}+\frac{\partial^{2}\theta }{\partial z^{2}})\\ C,I,D,E,C_{1}:T=T_{\infty }\\ B,F,G,H:-\lambda_{1}\frac{\partial T}{\partial n_{i}}=\alpha (T_{i}-T_{0})\\ A:\theta =f(x,y,z,t) \end{array}$$

When a workpiece is cut out, the tool is surrounded by air, resulting in it cooling down. The boundary conditions for the temperature field in this case were:(48)$$\{\begin{array}{l} \frac{\partial \theta }{\partial t}=\frac{\lambda }{c\cdot \rho }(\frac{\partial^{2}\theta }{\partial x^{2}}+\frac{\partial^{2}\theta }{\partial y^{2}}+\frac{\partial^{2}\theta }{\partial z^{2}})\\ C,I,D,E,C_{1}:T=T_{\infty }\\ B,F,G,H,A:-\lambda_{2}\frac{\partial T}{\partial n_{i}}=\alpha (T_{i}-T_{0}) \end{array}$$
where *T*_∞_ represents the tool’s temperature when it was stable. As the tool was clamped to the tool bar, this temperature is assumed to be room temperature. ni represents the direction of the outside normal of the rake face, *λ*_1_ is the tool’s thermal conductivity, *λ*_2_ is the thermal conductivity of the air, *ρ* represents the density of the cutting tool material, *c* is the specific heat capacity of the cutting tool, *α*_i_ is the convective heat conduction coefficient, and *θ* represents the tool’s heat density function.

### 4.3. Temperature Field Simulation

Prior to simulating the temperature field, the material properties of the micro-textured ball-end milling tool had to be added. The thermal conductivity of the tool was 75.4 W/(m·°C), its density was 14,700 kg/m^3^, its specific heat capacity was 470 J/(kg·°C), and the Poisson ratio was 0.3. With these details in place, the boundary conditions and heat density function could be applied to each surface to solve the calculation of the temperature field.

The simulation results are shown in Figure 11. It can be seen from these that, in the process of cutting titanium alloy, the friction between the cutting tool and the chip generates heat and the temperature of the tool surface gradually increases. This is because these two surfaces are in constant contact so, from the point of cutting-in to the point of cutting-out, the temperature will continually increase, reaching its highest value just as the chip is about to be cut out. It can also be seen from the simulation results that the maximum temperature was mainly concentrated on the rake face, the tool-chip contact area and the flank face near the cutting edge. This is because, during the cutting process, the principal source of heat was the friction between the tool and the workpiece in the contact area. As the contact area was small, the specific heat of the cutting tool was restricted, so heat conduction was also limited and only happened over a relatively short time. This meant that the temperature remained very high in the tool/chip contact area. The heat on the flank face was mainly produced by friction between the flank face and the surface of the workpiece. Adhesive wear was therefore most likely occurred in these two areas. As a consequence of the shape of the workpiece, the tool nose was not involved in the cutting, so the temperature at the tip of the tool was not very high. In the figure, the temperature gradient for the surface of the cutting tool is mostly uniform.

### 4.4. Thermal Stress Simulation

The transient temperature field simulation results were applied as load boundary conditions for a thermal stress simulation of the tool-chip contact area. At the point of loading, the reference temperature was set at 28 °C. Once the boundary conditions for displacement and load had been added, a finite element model for the thermal stress simulation was developed. Figure 12 shows the simulation results. Figure 12a is the equivalent stress nephogram of the tool and Figure 12b is the equivalent displacement nephogram of the tool.

From the equivalent stress nephogram, it can be seen that the thermal stress produced by the cutting temperature was mainly distributed in the tool-chip contact area and near the cutting edge on the flank face. Note that the thermal stress was at its largest along the cutting edge, measuring 2.0782 × 106 Pa. Moving away from the cutting edge, there was a decreasing trend along the contact length. The minimum thermal stress was 2.3091 × 105 Pa. As a result of the effects of the temperature field, the tool experienced heat expansion and micro-deformations appeared on the surface as the temperature increased. Thus, in the equivalent displacement nephogram, the maximum deformation is mainly concentrated at the cutting edge where the tool is in contact with the workpiece. The maximum value here was 1.9445 × 10^−9^ m. Moving away from the cutting edge, along the length or the width of the tool-chip contact area, the amount of deformation decreased. The minimum deformation caused by thermal stress was 2.1605 × 10^−10^ m. As micro-textured tools like this are subjected to alternating thermal stress at the cut-in and cut-out points, this can cause a concentration of stress, which often results in tiny cracks. Additionally, because of the difficulty of machining titanium alloy and its adhesive character, the tool-chip contact area and the cutting edge/workpiece contact area often experience felted wearing and thermal stress-induced breakage of the tool edge can occur during the cutting process.

## 5. Conclusions

(1) In this paper, we have analyzed the heating conditions that arise during the milling of titanium alloy with micro-textured ball-end milling tools. Scholars have never studied the thermo-mechanical coupling behavior of micro-textured ball-end milling cutters. This analysis was facilitated by simulation of the tool’s temperature field. The simulation results have shown that the rake face and the tool-chip contact area are particularly prone to heating and wear, whilst the temperature at the tool nose is less affected and the temperature gradient for the overall surface of the cutting tool is mostly uniform. This research can serve as a basis for further study of the thermo-mechanical coupling behavior of micro-textured ball-end milling tools.

(2) Through thermal stress simulation, we found that adhesive wear is especially likely to occur in the tool-chip contact area and on the flank face. Here, too, heat-related stress is not so obvious at the tip of the tool and the temperature distribution gradient across the tool in general is more or less uniform. The maximum value for thermal stress is on the cutting edge. Here it had a value of 2.0782 × 106 Pa. Thermal stress decreases as one moves away from the cutting edge along the length of the tool-chip contact area. The maximum deformation of the tool is mainly concentrated on the cutting edge where it is in contact with the workpiece. Here it had a maximum value of 1.9445 × 10^−9^ m. Once again, this decreases as one moves away from the cutting edge along the length and width of the tool-chip contact area.

## Figures and Tables

**Figure 1 micromachines-09-00672-f001:**
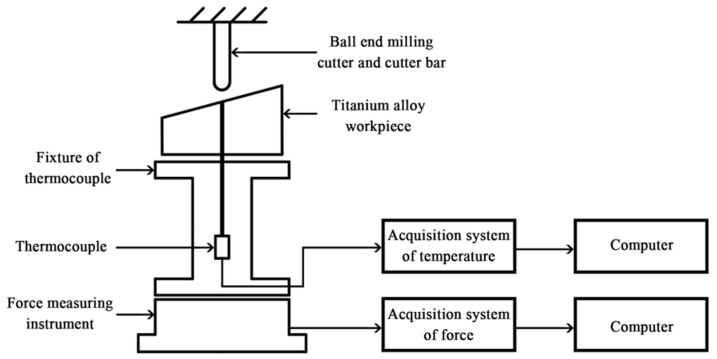
Approach to measuring the milling temperature.

**Figure 2 micromachines-09-00672-f002:**
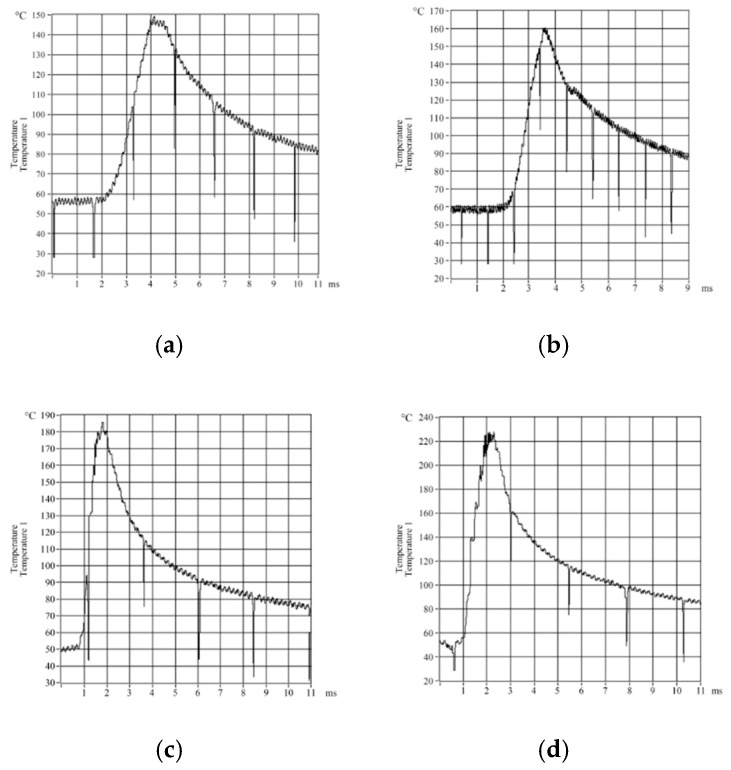
Curves for changes in milling temperature over time at different cutting speeds. (**a**) *v_c_* = 120 m/min; (**b**) *v_c_* = 140 m/min; (**c**) *v_c_* = 120 m/min; (**d**) *v_c_* = 140 m/min.

**Figure 3 micromachines-09-00672-f003:**
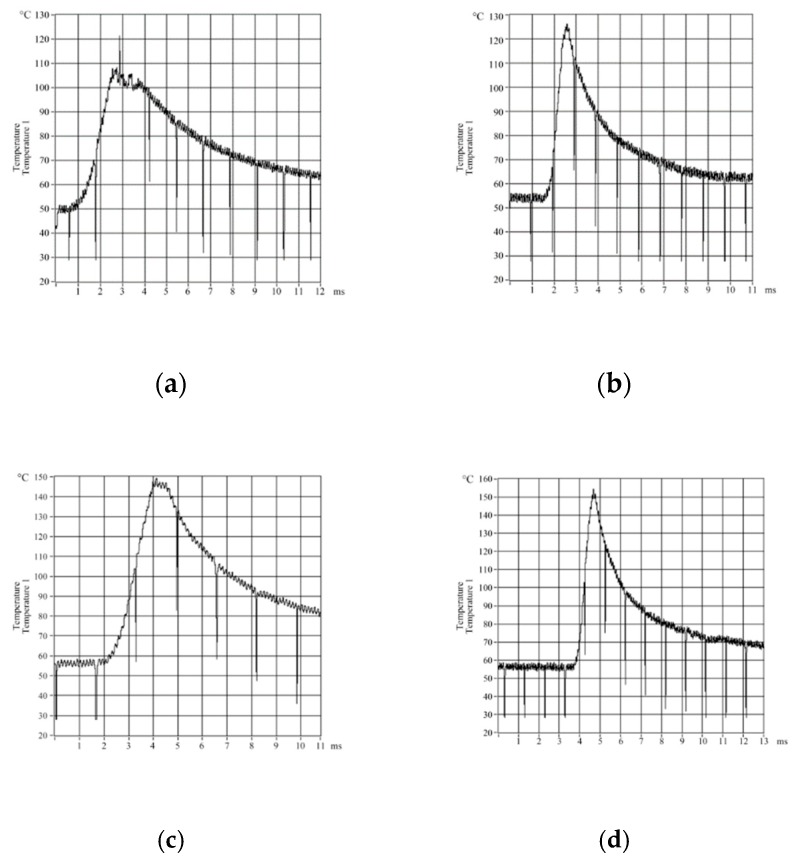
Curves for changes of milling temperature over time at different feed rates. (**a**) *f* = 0.04 mm/z; (**b**) *f* = 0.06 mm/z; (**c**) *f* = 0.08 mm/min; (**d**) *f* = 0.10 mm/z.

**Figure 4 micromachines-09-00672-f004:**
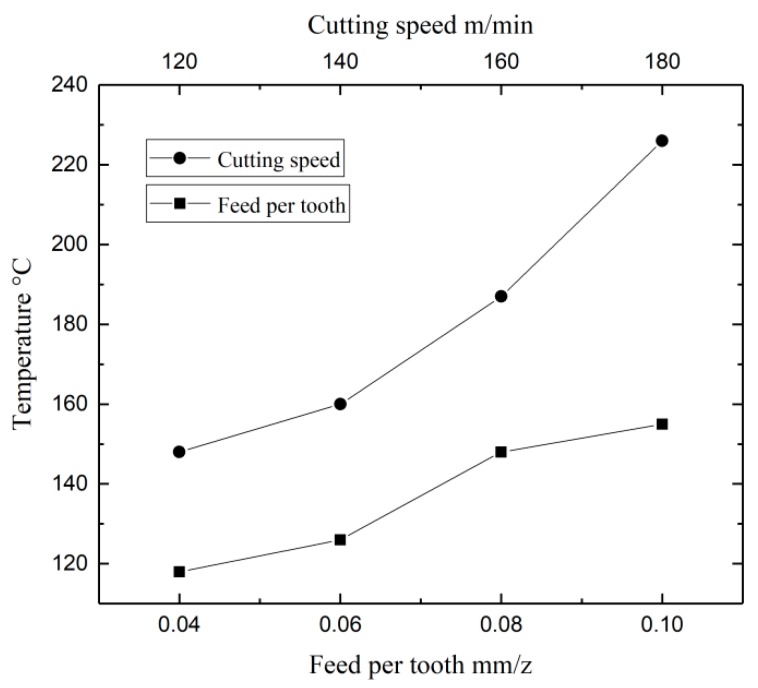
Changes in cutting temperature in relation to changes in the cutting parameters.

**Figure 5 micromachines-09-00672-f005:**
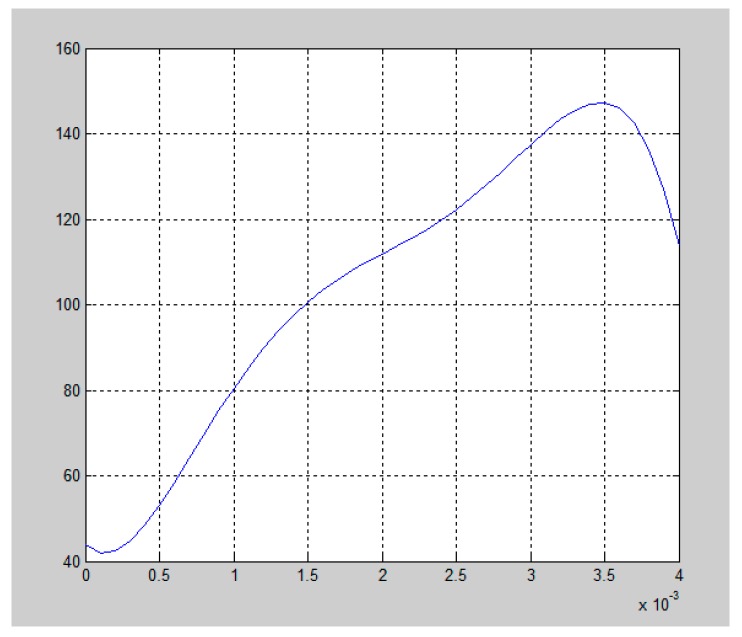
Curve for changes in milling temperature over time.

**Figure 6 micromachines-09-00672-f006:**
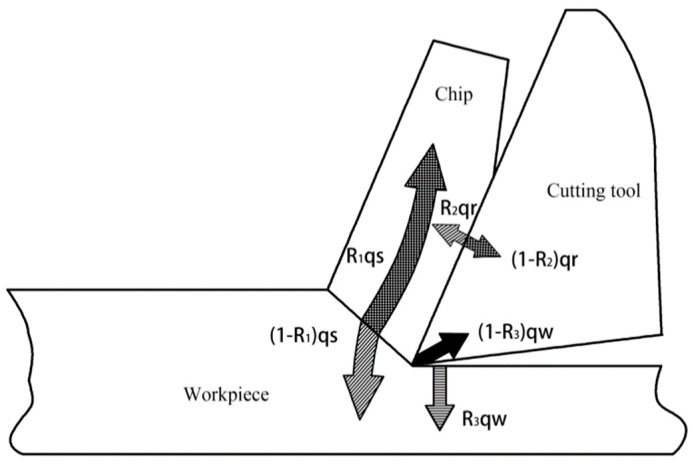
Generation and conduction of heat during cutting.

**Figure 7 micromachines-09-00672-f007:**
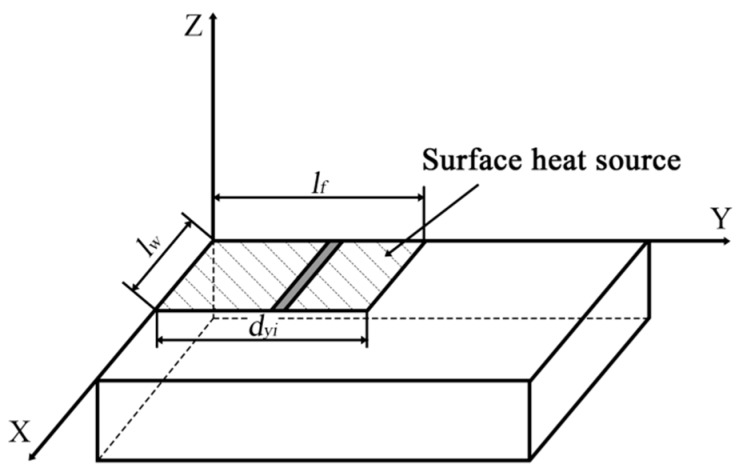
Schematic diagram of the surface heat source.

**Figure 8 micromachines-09-00672-f008:**
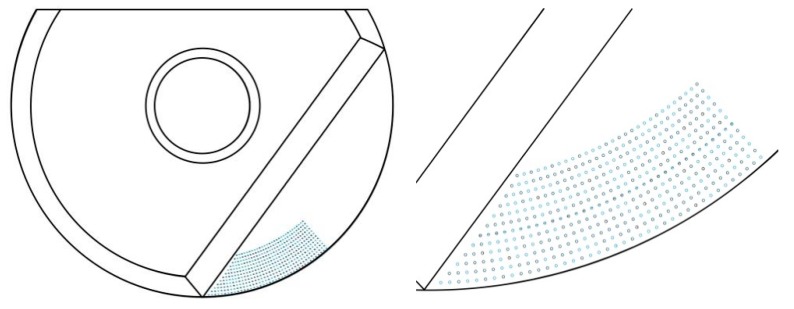
Blade model.

**Figure 9 micromachines-09-00672-f009:**
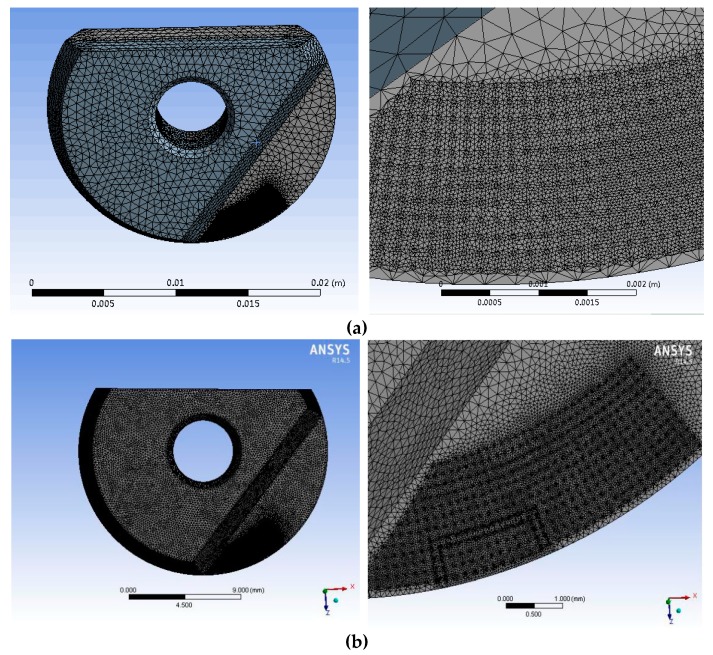
Mesh generation. (**a**) The mesh before optimization. (**b**) The optimized mesh.

**Figure 10 micromachines-09-00672-f010:**
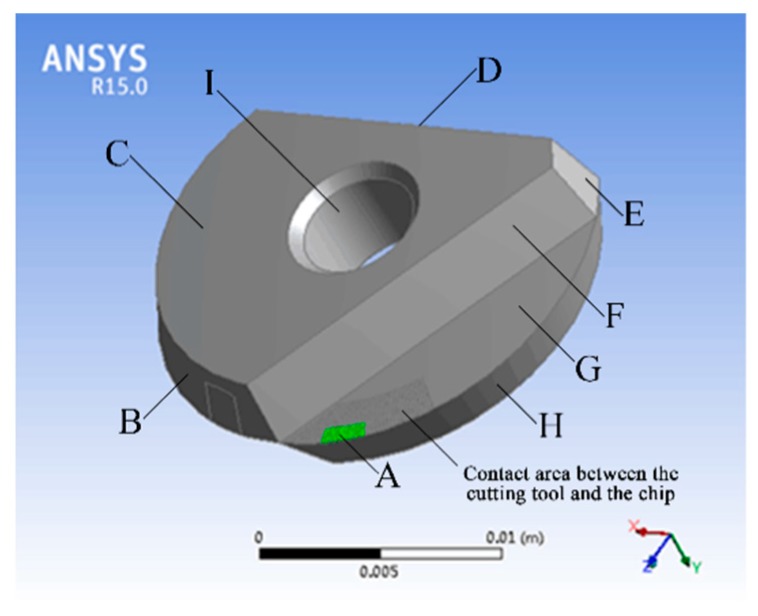
Overall depiction of the micro-texture ball-end milling tool.

**Figure 11 micromachines-09-00672-f011:**
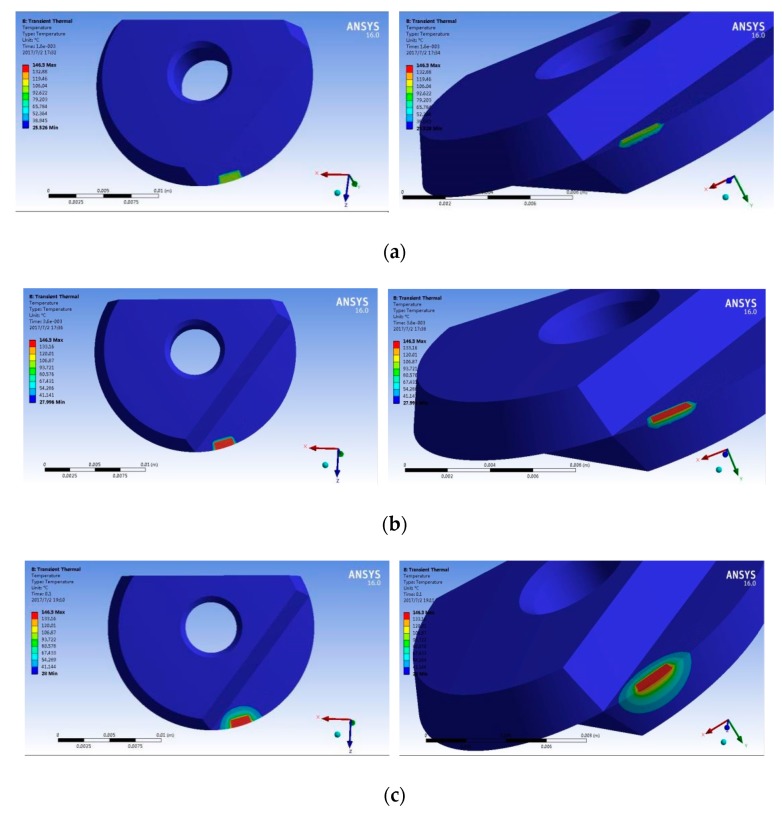
Changes in the temperature field over time. (**a**) t = 0.0016 s; (**b**) t = 0.0036 s; (**c**) t = 0.1 s.

**Figure 12 micromachines-09-00672-f012:**
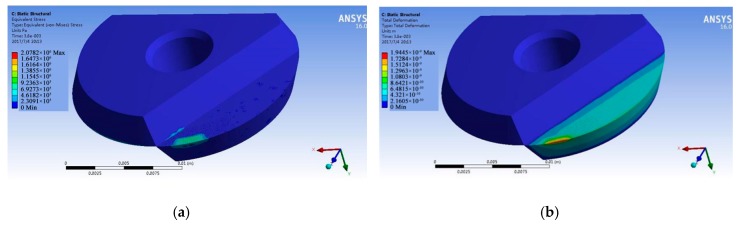
Nephograms showing the simulation results for thermal stress. (**a**) Equivalent stress; (**b**) equivalent displacement.

**Table 1 micromachines-09-00672-t001:** Single factor experiment for cutting titanium alloy.

	Cutting Parameter	Cutting Speed *v_c_* (m/min)	Cutting Depth *a_p_* (mm)	Feed Rate *f* (mm/z)
Number				
1		120	0.70	0.04
2				0.06
3				0.08
4				0.10
5		120	0.70	0.08
6		140		
7		160		
8		180		

**Table 2 micromachines-09-00672-t002:** Milling temperature data acquired across the milling cycle.

Acquisition Time within the Milling Cycle (s)	Milling Temperature (°C)
0.0004	48.693
0.0008	68.425
0.0012	91.244
0.0016	103.481
0.002	111.185
0.0024	118.475
0.0028	133.674
0.0032	141.586
0.0036	146.301

**Table 3 micromachines-09-00672-t003:** Dimensional analysis of the factors affecting the average temperature *θ*.

Variable	Dimension
The average temperature *θ*	$\bar{\theta }$
Cutting speed *v_c_*	*Lt* ^−1^
Cutting depth *a_c_*	*L*
Heat capacity of volume *ρc* and thermal conductivity coefficient *λ*	*M* ^2^ *t* ^−5^ $\cdot \bar{\theta }$ ^−2^
Specific cutting energy *u*	*ML*^−1^·*t*^−2^

**Table 4 micromachines-09-00672-t004:** Dimensional analysis of the factors that affect the average temperature in the tool-chip contact area.

Variable	Dimensional Unit
Cutting speed *v_c_*	m/s
Feed engagement *a_f_*	m/z
Cutting depth *a_c_*	m
Cutting width *a_e_*	m
Tool diameter *D*	m
Thermal conductivity of the workpiece material *λ*	W/(m·*K*）
Workpiece material density *ρ*	kg/m^3^
Specific cutting energy of the workpiece material *c*	J/(kg·*K*)

**Table 5 micromachines-09-00672-t005:** Parameters and dimensional analysis relating to the cutting of the workpiece material.

Variable	Numerical Value	Dimensional Unit
Cutting speed *v_c_*	2	m/s
Feed engagement *a_f_*	0.00008	m/z
Cutting depth *a_c_*	0.00007	m
Cutting width *a_e_*	0.00005	m
Tool diameter *D*	0.02	m
Tensile strength	0.895	GPa
Thermal conductivity of the workpiece material *λ*	15.24	W/(m·*K*)
Workpiece material density *ρ*	4.5 × 10^3^	kg/m^3^
Specific heat capacity of the workpiece material *c*	611	J/(kg·*K*)

**Table 6 micromachines-09-00672-t006:** Material parameters for the tool.

Density kg/m^3^	Thermal Conductivity (W/(m·°C))	Coefficient of Thermal Expansion *α* (×10^−6^·°C^−1^)	Modulus of Elasticity E(GPa)	Poisson Ratio	Specific Heat Capacity C(J/(kg·°C))	Melting Point °C	Boiling Point °C
14,700	75.4	4.5	540	0.3	470	2780	6000

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
