# Peer review of "Temperature Field of Tool Engaged Cutting Zone for Milling of Titanium Alloy with Ball-End Milling"

_micromachines, 2018, doi:10.3390/mi9120672_

Round 1

Reviewer 1 Report

An excellent paper on the micromachining of materials using ball ended cutting tools.  Some minor checking of the English grammar is required.

Author Response

 We appreciate the thorough reviews and positive comments provided by both yourself and the reviewers on our submitted paper. According to these objective and professional suggestions, careful revisions have been made.

Q: Some minor checking of the English grammar is required.

A:Thank you very much! I have check the Eglish grammar and I've asked professionals to revise my paper.

Reviewer 2 Report

This paper describes thermal analysis of tool-chip contact area supported by temperature measurment during ball endmil process. The main purpose was to derive a heat flux density function which will be frequently utilized to design tool structures.

It's very impressed that a dimensional analysis for the factors affecting the average temperature was adopted, which produces contributions of heat capacity, specific cutting energy, cutting speed, cutting depth. Also, the heat density function was derived based on the cutting force.

Based on the simulation, the authors found that the adhesive wear was likely to occur in the tool-chip contact area and the flank face.

comment 1 : The caption of Fig.6 should be corrected.

comment 2 : The title makes confusing to understand the context, hence it should be modified, such as 'Temperature field of tool engaged cutting zone for milling of titanium alloy with ball-end milling'   

Author Response

We appreciate the thorough reviews and positive comments provided by both yourself and the reviewers on our submitted paper. According to these objective and professional suggestions, careful revisions have been made.

Q:The caption of Fig.6 should be corrected.

A: Curves for changes of milling temperature over time at different feed rates

Q:The title makes confusing to understand the context, hence it should be modified, such as 'Temperature field of tool engaged cutting zone for milling of titanium alloy with ball-end milling'   

A: I have revised.

Reviewer 3 Report

The paper topic is interesting and paper can be published but only after the above suggestions:

Weak points of this manuscript are introduction and conclusions. Those parts have to rewrite.

Abstract: The abstract is expected to include a brief digest of the research, that is, new methods, results, concepts, and conclusions only. The abstract needs to be more focused and achievements needs mentioned clearly. At the moment abstract is more like an introduction than abstract. Please add some information from the conclusion (quantifications).

Introduction based on old references – youngest paper has 4 years, rest sometimes more than 20-30 years. Introduction is expected to have an extensive literature review followed by an in-depth and critical analysis of the state of the art. Each one of the cited references within the body of the paper should be discussed individually and explicitly to demonstrate their significance to the study. If you avoided reference overkill/run-on, i.e. do not use more than 3 references per sentence. If you need to use more, make sure you state the key relevant idea of each reference. References section should be extensive about information connecting with precision manufacturing and ball end mills research. Authors cited papers only from one side of world and ignore others research. I suggest add information to better describe what other researchers have done in this area for example prof Szymon Wojciechowski who published very similar research – you should compare your result with him works. I suggest add important and new articles from this field:

Study on metrological relations between instant tool displacements and surface roughness during precise ball end milling, Measurement, 129, pp. 686-694 (2018).

Optimisation of machining parameters during ball end milling of hardened steel with various surface inclinations, Measurement, 111, pp. 18 – 28 (2017).

Modeling of cutter displacements during ball end milling of inclined surfaces, Archives of Civil and Mechanical Engineering, 15 (4), pp. 798 – 805 (2015).

Figs. 1a, 2, 3 do not contribute anything important to the paper and should be removed.

All variables parameters should be written in italic style.

Cutting speed is Vc – not V

In Table 1 authors wrote 0.001 seconds. The measurement device (typical) has a lower interaction. So how reliable are the data presented? Not to three decimal places!

The discussion is shallow and needs more details, the observations and future trends. This chapter should be connected with others published papers.

Conclusion are simplistic;  Please try to emphasize your novelty, put some quantifications, and comment on the limitations.

Author Response

We appreciate the thorough reviews and positive comments provided by both yourself and the reviewers on our submitted paper. According to these objective and professional suggestions, careful revisions have been made.

Q:Weak points of this manuscript are introduction and conclusions. Those parts have to rewrite.

A:I have rewrite it.

Q: Introduction based on old references – youngest paper has 4 years, rest sometimes more than 20-30 years. Introduction is expected to have an extensive literature review followed by an in-depth and critical analysis of the state of the art. Each one of the cited references within the body of the paper should be discussed individually and explicitly to demonstrate their significance to the study. If you avoided reference overkill/run-on, i.e. do not use more than 3 references per sentence. If you need to use more, make sure you state the key relevant idea of each reference. References section should be extensive about information connecting with precision manufacturing and ball end mills research. Authors cited papers only from one side of world and ignore others research. I suggest add information to better describe what other researchers have done in this area for example prof Szymon Wojciechowski who published very similar research – you should compare your result with him works.

A:I have added them.

Q: All variables parameters should be written in italic style.

Cutting speed is Vc – not V.

A: I have corrected it.

Q: In Table 1 authors wrote 0.001 seconds. The measurement device (typical) has a lower interaction. So how reliable are the data presented? Not to three decimal places!

A:The test device is connected to the computer and the data is output directly by the computer.

Q:The discussion is shallow and needs more details, the observations and future trends. This chapter should be connected with others published papers.

A:.I have already written it in my paper.

Q:Conclusion are simplistic;  Please try to emphasize your novelty, put some quantifications, and comment on the limitations.

A:I have corrected it.

Reviewer 4 Report

Comments:

1) What type of experimental design are you using? "We adopted a single factor approach for the experiment."

2) How the factors and levels are selected? For instance, why the authors selected just one value for ap? Is it not important for temperature?

3) Figure 13 has letters in ¿Chinese? Please, correct it.

4) Quality of images, especially the last ones, should be improved in order to improve readability.

5) The authors are presenting experimental and simulated results. It would be interesting to have a section on the paper comparing these two approaches. 

Author Response

We appreciate the thorough reviews and positive comments provided by both yourself and the reviewers on our submitted paper. According to these objective and professional suggestions, careful revisions have been made.

Q:What type of experimental design are you using? "We adopted a single factor approach for the experiment."

A: Yes, We adopted a single factor approach for the experiment..

Q:How the factors and levels are selected? For instance, why the authors selected just one value for ap? Is it not important for temperature?

A:We adopted a single factor approach for the experiment. The number of the factors is 2 and  the number of the levels is 4. In other In another paper, the empirical formula of milling force is derived from the experiment of the above parameters. In order to obtain the thermo-mechanical coupling field of micro-textured ball-end milling cutter, the same parameters are needed, so the cutting depth is chosen.

Q:Figure 13 has letters in ¿Chinese? Please, correct it.

A:  I have corrected it.

Q:Quality of images, especially the last ones, should be improved in order to improve readability.

A:I've magnified it.

Q:The authors are presenting experimental and simulated results. It would be interesting to have a section on the paper comparing these two approaches.

A: Based on the experimental data, the boundary conditions of simulation are obtained, so they are not comparative.

Round 2

Reviewer 3 Report

Paper is ready for publication

Author Response

Thank you very much!

Reviewer 4 Report

Thanks for considering my comments. I should continue with two of them because I haven't seen an "adequate" answer. So:

1) Ok! You "adopted a single factor approach for the experiment". But why? Can you explain the benefits of this approach? Why not using another one? I think that readers could benefit from some insights on this selection.

2) How the factors and levels are selected? For instance, why the authors selected just one value for ap? Is it not important for temperature?

I'm not happy with this answer. I think that instead of referring to other "another paper", you should state in this one why you are selecting these parameters. Why are you only working with 1 value for depth of cut? Why selecting these specific values of feed rate, cutting speed? I miss an explanation in the paper.

Besides, I'm answering "must be improved" to "Does the introduction provide sufficient background and include all relevant references?". I think that you could work a little bit on that. You are presenting just 19 references. Please, try to include some of the latest works.

Author Response

 1. Ok! You "adopted a single factor approach for the experiment". But why? Can you explain the benefits of this approach? Why not using another one? I think that readers could benefit from some insights on this selection.

A:In this paper, the influence of milling parameters on cutting temperature is analyzed, so single factor test is adopted instead of orthogonal test.

2.How the factors and levels are selected? For instance, why the authors selected just one value for ap? Is it not important for temperature?

A: Under finishing conditions of high speed milling, ap has little effect on cutting temperature. Therefore, in combination with previous studies, this parameter is selected.In previous studies, the cutter-chip contact area was determined, which was the closest to the cutter contact area under this parameter. Therefore, this parameter was chosen. In order to ensure the unity of the study, the cutting temperature under this parameter was studied. Because another article was reviewed, it could not be cited.

3. Besides, I'm answering "must be improved" to "Does the introduction provide sufficient background and include all relevant references?". I think that you could work a little bit on that. You are presenting just 19 references. Please, try to include some of the latest works.

At present, few people do this research. A few years ago, most scholars studied turning tools, but the turning and milling processes are very different, so the reference content is limited. There is no research on temperature field of micro-textured ball-end milling cutter, so there are few references and few studies in recent years.

     If you have any questions, please do not hesitate to contact me. Thank you very much!